# NODE FEATURE EXTRACTION BY SELF-SUPERVISED MULTI-SCALE NEIGHBORHOOD PREDICTION

**Eli Chien**[*]
University of Illinois Urbana-Champaign, USA
`ichien3@illinois.edu`

**Wei-Cheng Chang**
Amazon, USA
`chanweic@amazon.com`

**Cho-Jui Hsieh**
University of California, Los Angeles, USA
`chohsieh@cs.ucla.edu`

**Hsiang-Fu Yu, Jiong Zhang**
Amazon, USA
`{hsiangfu,jiongz}@amazon.com`

**Olgica Milenkovic**
University of Illinois Urbana-Champaign, USA
`milenkov@illinois.edu`

**Inderjit S. Dhillon**
Amazon, USA
`isd@amazon.com`

## ABSTRACT

Learning on graphs has attracted significant attention in the learning community due to numerous real-world applications. In particular, graph neural networks (GNNs), which take *numerical* node features and graph structure as inputs, have been shown to achieve state-of-the-art performance on various graph-related learning tasks. Recent works exploring the correlation between numerical node features and graph structure via self-supervised learning have paved the way for further performance improvements of GNNs. However, methods used for extracting numerical node features from *raw data* are still *graph-agnostic* within standard GNN pipelines. This practice is sub-optimal as it prevents one from fully utilizing potential correlations between graph topology and node attributes. To mitigate this issue, we propose a new self-supervised learning framework, Graph Information Aided Node feature exTraction (GIANT). GIANT makes use of the eXtreme Multi-label Classification (XMC) formalism, which is crucial for fine-tuning the language model based on graph information, and scales to large datasets. We also provide a theoretical analysis that justifies the use of XMC over link prediction and motivates integrating XR-Transformers, a powerful method for solving XMC problems, into the GIANT framework. We demonstrate the superior performance of GIANT over the standard GNN pipeline on Open Graph Benchmark datasets: For example, we improve the accuracy of the top-ranked method GAMLP from 68.25% to 69.67%, SGC from 63.29% to 66.10% and MLP from 47.24% to 61.10% on the ogbn-papers100M dataset by leveraging GIANT. Our implementation is public available[1].

## 1 INTRODUCTION

The ubiquity of graph-structured data and its importance in solving various real-world problems such as node and graph classification have made graph-centered machine learning an important research area (Lü & Zhou, 2011; Shervashidze et al., 2011; Zhu, 2005). Graph neural networks (GNNs) offer state-of-the-art performance on many graph learning tasks and have by now become a standard methodology in the field (Kipf & Welling, 2017; Hamilton et al., 2017; Velickovic et al., 2018; Chien et al., 2020). In most such studies, GNNs take graphs with *numerical node attributes* as inputs and train them with task-specific labels.

Recent research has shown that self-supervised learning (SSL) leads to performance improvements in many applications, including graph learning, natural language processing and computer vision.

---

[*]This work was done during Eli Chien's internship at Amazon, USA.
[1]`https://github.com/amzn/pecos/tree/mainline/examples/giant-xrt`

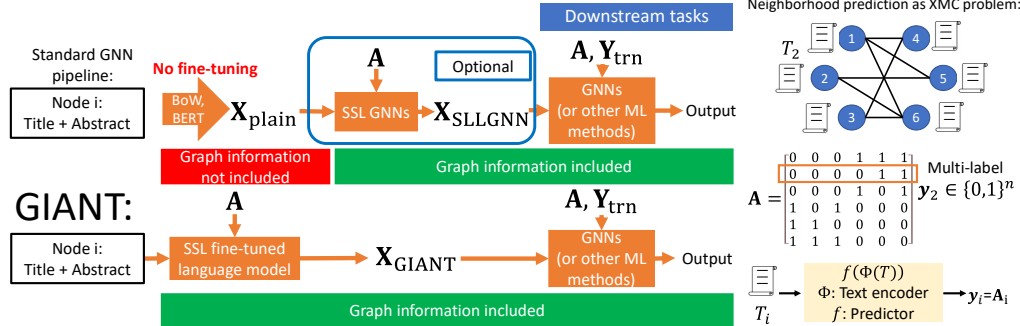

Figure 1: Left: Illustration of the standard GNN pipeline and our GIANT framework. Note that only the language model (i.e., the Transformers) in GIANT can use the correlation between graph topology and raw text. Nevertheless, GIANT can work with other types of input data formats, such as images and audio; the study of these models is deferred to future work. Right: Illustration of the connection between our neighborhood prediction and the XMC problem. We use graph information to self-supervise fine-tuning of the text encoder $\Phi$ (i.e., the Transformer) in our neighborhood prediction problem. The resulting fine-tuned text encoder is then used to generate numerical node features $\mathbf{X}_{\text{GIANT}}$ for use in downstream tasks (see also Figure 2 and the description in Section 3).

Several SSL approaches have also been successfully used with GNNs (Hu et al., 2020b; You et al., 2018; 2020; Hu et al., 2020c; Velickovic et al., 2019; Kipf & Welling, 2016; Deng et al., 2020). The common idea behind these works is to explore the correlated information provided by the *numerical* node features and graph topology, which can lead to improved node representations and GNN initialization. However, one critical yet neglected issue in the current graph learning literature is how to actually obtain *the numerical node features from raw data* such as text, images and audio signals. As an example, when dealing with raw text features, the standard approach is to apply *graph-agnostic* methods such as bag-of-words, word2vec (Mikolov et al., 2013) or pre-trained BERT (Devlin et al., 2019) (As a further example, raw texts of product descriptions are used to construct node features via the bag-of-words model for benchmarking GNNs on the ogbn-products dataset (Hu et al., 2020a; Chiang et al., 2019)). The pre-trained BERT language model, as well as convolutional neural networks (CNNs) (Goyal et al., 2019; Kolesnikov et al., 2019), produce numerical features that can significantly improve the performance of various downstream learners (Devlin et al., 2019). Still, none of these works leverage graph information for actual self-supervision. Clearly, using graph-agnostic methods to extract numerical features is sub-optimal, as correlations between the graph topology and raw features are ignored.

Motivated by the recent success of SSL approaches for GNNs, we propose GIANT, an SSL framework that resolves the aforementioned issue of graph-agnostic feature extraction in the standard GNN learning pipeline. Our framework takes raw node attributes and generates numerical node features with graph-structured self-supervision. To integrate the graph topology information into language models such as BERT, we also propose a novel SSL task termed *neighborhood prediction*, which works for both homophilous and heterophilous graphs, and establish connections between neighborhood prediction and the eXtreme Multi-label Classification (XMC) problem (Shen et al., 2020; Yu et al., 2022; Chang et al., 2020b). Roughly speaking, the neighborhood of each node can be encoded using binary multi-labels (indicating whether a node is a neighbor or not) and the BERT model is fine-tuned by successively improving the predicted neighborhoods. This approach allows us to not only leverage the advanced solvers for the XMC problem and address the issue of graph-agnostic feature extraction, but also to perform a theoretical study of the XMC problem and determine its importance in the context of graph-guided SSL.

Throughout the work, we focus on raw texts as these are the most common data used for large-scale graph benchmarking. Examples include titles/abstracts in citation networks and product descriptions in co-purchase networks. To solve our proposed self-supervised XMC task, we adopt the state-of-the-art XR-Transformer method (Zhang et al., 2021a). By using the encoder from the XR-Transformer pre-trained with GIANT, we obtain informative numerical node features which consistently boost the performance of GNNs on downstream tasks.

Notably, GIANT significantly improves state-of-the-art methods for node classification tasks described on the Open Graph Benchmark (OGB) (Hu et al., 2020a) leaderboard on three large-scale graph datasets, with absolute improvements in accuracy roughly $1.5\%$ for the first-ranked methods, $3\%$ for standard GNNs and $14\%$ for multilayer perceptron (MLP). GIANT coupled with XR-Transformer is also highly scalable and can be combined with other downstream learning methods.

Our contributions may be summarized as follows.

**1.** We identify the issue of graph-agnostic feature extraction in standard GNN pipelines and propose a new GIANT self-supervised framework as a solution to the problem.

**2.** We introduce a new approach to extract numerical features by graph information based on the idea of *neighborhood prediction.* The gist of the approach is to use neighborhood prediction within a language model such as BERT to guide the process of fine-tuning the features. Unlike link-prediction, neighborhood prediction resolves problems associated with heterophilic graphs.

**3.** We establish pertinent connections between neighborhood prediction and the XMC problem by noting that neighborhoods of individual nodes can be encoded by binary vectors which may be interpreted as multi-labels. This allows for performing neighborhood prediction via XR-Transformers, especially designed to solve XMC problems at scale.

**4.** We demonstrate through extensive experiments that GIANT consistently improves the performance of tested GNNs on downstream tasks by large margins. We also report new state-of-the-art results on the OGB leaderboard, including absolute improvements in accuracy roughly $1.5\%$ compared to the top-ranked method, $3\%$ for standard GNNs and $14\%$ for multilayer perceptron (MLP). More precisely, we improve the accuracy of the top-ranked method GAMLP (Zhang et al., 2021b) from $68.25\%$ to $69.67\%$, SGC (Wu et al., 2019) from $63.29\%$ to $66.10\%$ and MLP from $47.24\%$ to $61.10\%$ on the ogbn-papers100M dataset.

**5.** We present a new theoretical analysis that verifies the benefits of key components in XR-Transformers on our neighborhood prediction task. This analysis also further improves our understanding of XR-Transformers and the XMC problem.

Due to the space limitation, all proofs are deferred to the Appendix.

## 2 BACKGROUND AND RELATED WORK

**General notation.** Throughout the paper, we use bold capital letters such as $\mathbf{A}$ to denote matrices. We use $\mathbf{A}_i$ for the $i$-th row of the matrix and $\mathbf{A}_{ij}$ for its entry in row $i$ and column $j$. We reserve bold lowercase letters such as $\mathbf{a}$ for vectors. The symbol $\mathbf{I}$ denotes the identity matrix while $\mathbf{1}$ denotes the all-ones vector. We use $o(\cdot), O(\cdot), \omega(\cdot), \Theta(\cdot)$ in the standard manner.

**SSL in GNNs.** SSL is a topic of substantial interest due to its potential for improving the performance of GNNs on various tasks. Exploiting the correlation between node features and the graph structure is known to lead to better node representations or GNN initialization (Hu et al., 2020b; You et al., 2018; 2020; Hu et al., 2020c). Several methods have been proposed for improving node representations, including (variational) graph autoencoders (Kipf & Welling, 2016), Deep Graph Infomax (Velickovic et al., 2019) and GraphZoom (Deng et al., 2020). For more information, the interested reader is referred to a survey of SSL GNNs (Xie et al., 2021). While these methods can be used as SSL modules in GNNs (Figure 1), it is clear that they do not solve the described graph agnostics issue in the standard GNN pipeline. Furthermore, as the above described SSL GNNs modules and other pre-processing and post-processing methods for GNNs such as C&S (Huang et al., 2021) and FLAG (Kong et al., 2020) in general improve graph learners, it is worth pointing out that they can be naturally be integrated into the GIANT framework. This topic is left as a future work.

**The XMC problem, PECOS and XR-Transformer.** The XMC problem can be succinctly formulated as follows: We are given a training set $\{T_i, \mathbf{y}_i\}_{i=1}^n$, where $T_i \in \mathcal{D}$ is the $i$th input text instance and $\mathbf{y}_i \in \{0,1\}^L$ is the target multi-label from an extremely large collection of labels. The goal is to learn a function $f : \mathcal{D} \times [L] \mapsto \mathbb{R}$, where $f(T, l)$ captures the relevance between the input text $T$ and the label $l$. The XMC problem is of importance in many real-world applications (Jiang et al., 2021; Ye et al., 2020): For example, in E-commerce dynamic search advertising, XMC arises when trying to find a "good" mapping from items to bid queries on the market (Prabhu et al., 2018; Prabhu & Varma, 2014). In open-domain question answering, XMC problems arise when trying to map ques-

tions to "evidence" passages containing the answers (Chang et al., 2020a; Lee et al., 2019). Many methods for the XMC problem leverage hierarchical clustering approaches for labels (Prabhu et al., 2018; You et al., 2019). This organizational structure allows one to handle potentially enormous numbers of labels, such as used by PECOS (Yu et al., 2022). The key is to take advantage of the correlations among labels within the hierarchical clustering. In our approach, we observe that the multi-labels correspond to neighborhoods of nodes in the given graph. Neighborhoods have to be predicted using the textual information in order to best match the a priori given graph topology. We use the state-of-the-art XR-Transformer (Zhang et al., 2021a) method for solving the XMC problem to achieve this goal. The high-level idea is to first cluster the output labels, and then learn the instance-to-cluster "matchers" (please refer to Figure 2). Note that many other methods have used PECOS (including XR-Transformers) for solving large-scale real-world learning problems (Etter et al., 2022; Liu et al., 2021; Chang et al., 2020b; Baharav et al., 2021; Chang et al., 2021; Yadav et al., 2021; Sen et al., 2021), but not in the context of self-supervised numerical feature extraction as done in our work.

**GNNs with raw text data.** It is conceptually possible to jointly train BERT and GNNs in an end-to-end fashion, which could potentially resolve the issue of being graph agnostic in the standard pipeline. However, the excessive model complexity of BERT makes such a combination practically prohibitive due to GPU memory limitations. Furthermore, it is nontrivial to train this combination of methods with arbitrary mini-batch sizes (Chiang et al., 2019; Zeng et al., 2020). In contrast, the XR-Transformer architecture naturally supports mini-batch training and scales well (Jiang et al., 2021). Hence, our GIANT method uses XR-Transformers instead of combinations of BERT and GNNs. To the best of our knowledge, we are aware of only one prior work that uses raw text inputs for node classification problem (Zhang et al., 2020), but it still follows the standard pipeline described in Figure 1. Some other works apply GNNs on texts and for document classification, where the actual graphs are constructed based on the raw text. This is clearly not the focus of this work (Yao et al., 2019; Huang et al., 2019; Zhang & Zhang, 2020; Liu et al., 2020).

## 3 METHODS

Our goal is to resolve the issue of graph-agnostic numerical feature extraction for standard GNN learning pipelines. Although our interest lies in raw text data, as already pointed out, the proposed methodology can be easily extended to account for other types of raw data and corresponding feature extraction methods.

To this end, consider a large-scale graph $G$ with node set $\mathcal{V} = \{1, 2, \ldots, n\}$ and adjacency matrix $\mathbf{A} \in \{0, 1\}^{n \times n}$. Each node $i$ is associated with some raw text, which we denote by $T_i$. The language model is treated as an encoder $\Phi$ that maps the raw text $T_i$ to numerical node feature $\mathbf{X}_i \in \mathbb{R}^d$. Key to our SSL approach is the task of *neighborhood prediction*, which aims to determine the neighborhood $\mathbf{A}_i$ from $T_i$. The neighborhood vector $\mathbf{A}_i$ can be viewed as a target multi-label $\mathbf{y}_i$ for node $i$, where we have $L = n$ labels. Hence, neighborhood prediction represents an instance of the XMC problem, which we solve by leveraging XR-Transformers. The trained encoder in an XR-Transformer generates informative numerical node features, which can then be used further in downstream tasks, the SSL GNNs module and for GNN pre-training.

**Detailed description regarding the use of XR-Transformers for Neighborhood Prediction.** The most straightforward instance of the XMC problem is the one-versus-all (OVA) model, which can be formalized as $f(T, l) = \mathbf{w}_l^T \Phi(T); \ l \in [L]$, where $\mathbf{W} = [\mathbf{w}_1, \ldots, \mathbf{w}_L] \in \mathbb{R}^{d \times L}$ are weight vectors and $\Phi : \mathcal{D} \mapsto \mathbb{R}^d$ is the encoder that maps $T$ to a $d$-dimensional feature vector. OVA can be a deterministic model such as bag-of-words, the Term Frequency-Inverse Document Frequency (TFIDF) model or some other model with learnable parameters, such as XLNet (Yang et al., 2019) and RoBERTa (Liu et al., 2019b). We choose to work with pre-trained BERT (Devlin et al., 2019). Also, one can change $\Phi$ according to the type of input data format (i.e., CNNs for images). Despite their simple formulation, it is known Chang et al. (2020b) that fine-tuning transformer models directly on large output spaces can be prohibitively complex. For neighborhood prediction, $L = n$, and the graphs encountered may have millions of nodes. Hence, we need a more scalable approach to training Transformers. As part of an XR-Transformer, one builds a hierarchical label clustering tree based on the label features $\mathbf{Z} \in \mathbb{R}^{L \times d}$; $\mathbf{Z}$ is based on Positive Instance Feature Aggregation

Figure 2: Illustration of the use of XR-Transformers. Step 1: Perform semantic hierarchical clustering of target labels (neighborhoods) to build a tree. Step 2: At each intermediate (internal node) level of the tree, fine-tune the Transformers for the XMC sub-problem that maps raw text of nodes to label clusters. Note that the results of higher levels are used to guide the Transformers at lower levels and hence improve their performance. The resulting Transformers are used as encoders that generate numerical node features from raw texts. Note that we can change the encoder (e.g., Transformer) to address other raw data formats such as images or audio signals.

(PIFA):

$$\mathbf{Z}_l = \frac{\mathbf{v}_l}{\|\mathbf{v}_l\|}, \text{ where } \mathbf{v}_l = \sum_{i: \mathbf{y}_{i,l}=1} \Psi(T_i), \ \forall l \in [L]. \tag{1}$$

Note that for neighborhood prediction, the above expression represents exactly one step of a graph convolution with node features $\Psi(T_i)$, followed by a norm normalization; here, $\Psi(\cdot)$ denotes some text vectorizer such as bag-of-words or TFIDF. In the next step, XR-Transformer uses balanced $k$-means to recursively partition label sets and generate the hierarchical label cluster tree in a top-down fashion. This step corresponds to Step 1 in Figure 2. Note that at each intermediate level, it learns a matcher to find the most relevant clusters, as illustrated in Step 2 of Figure 2. By leveraging the label hierarchy defined by the cluster tree, the XR-Transformer can train the model on multi-resolution objectives. Multi-resolution learning has been used in many different contexts, including computer vision (Lai et al., 2017; Karras et al., 2018; 2019; Pedersoli et al., 2015), meta-learning (Liu et al., 2019a), but has only recently been applied to the XMC problem as part of PECOS and XR-Transformers. For neighborhood prediction, multi-resolution amounts to generating a hierarchy of coarse-to-fine views of neighborhoods. The only line of work in self-supervised graph learning that somewhat resembles this approach is GraphZoom (Deng et al., 2020), in so far that it applies SSL on coarsened graphs. Nevertheless, the way in which we perform coarsening is substantially different; furthermore, GraphZoom still falls into the standard GNN pipeline category depicted in Figure 1.

## 4 THEORETICAL ANALYSIS

We also provide theoretical evidence in support of using each component of our proposed learning framework. First, we show that self-supervised neighborhood prediction is better suited to the task at hand than standard link prediction. More specifically, we show that the standard design criteria in self-supervised link prediction tasks are biased towards graph homophily assumptions (McPherson et al., 2001; Klicpera et al., 2018). In contrast, our self-supervised neighborhood prediction model works for both homophilic and heterophilic graphs. This universality property is crucial for the robustness of graph learning methods, especially in relationship to GNNs (Chien et al., 2020). Second, we demonstrate the benefits of using PIFA embeddings and clustering in XR-Transformers for graph-guided numerical feature extraction. Our analysis is based on the contextual stochastic block model (cSBM) (Deshpande et al., 2018), which was also used in Chien et al. (2020) for testing the GPR-GNN framework and in Baranwal et al. (2021) for establishing the utility of graph convolutions for node classification.

**Link versus neighborhood prediction.** One standard SSL task on graphs is link prediction, which aims to find an entry in the adjacency matrix according to

$$\mathbb{P}(\mathbf{A}_{ij} = 1) \propto \text{Similarity}(\Phi(T_i), \Phi(T_j)). \tag{2}$$

Here, the function Similarity$(\mathbf{x}, \mathbf{y})$ is a measure of similarity of two vectors, $\mathbf{x}$ and $\mathbf{y}$. The most frequently used choice for the function is the inner product of two input vectors followed by a sigmoid function. However, this type of design implicitly relies on the homophily assumption:

*Nodes with similar node representations are more likely to have links.* It has been shown in Pei et al. (2020); Chien et al. (2020); Zhu et al. (2020); Lim et al. (2021) that there are real-world graph datasets that violate the homophily assumption and on which many GNN architectures fail.

A simple example that shows how SSL link prediction may fail is presented in Figure 3. Nodes of the same color share the same features (these are for simplicity represented as numerical values). Clearly, no matter what encoder $\Phi$ we have, the similarity of node features for nodes of the same color is the highest. However, there is no edge between nodes of the same color, hence the standard methodology of link prediction based on homophily assumption fails to work for this simple heterophilous graph. In order to fix this issue, we use a different modeling assumption, stated below.

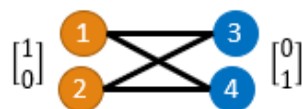

Figure 3: A counter-example for standard link prediction methodology.

**Assumption 4.1.** *Nodes with similar node features have similar "structural roles" in the graph. In our study, we equate "structure" with the 1-hop neighborhood of a node (i.e., the row of the adjacency matrix indexed by the underlying node).*

The above assumption is in alignment with our XMC problem assumptions, where nodes with a small perturbation in their raw text should be mapped to a similar multi-label. Our assumption is more general then the standard homophily assumption; it is also clear that there exists a perfect mapping from node features to their neighborhoods for the example in Figure 3. Hence, neighborhood prediction appears to be a more suitable SSL approach than SSL link prediction for graph-guided feature extraction.

**Analysis of key components in XR-Transformers.** In the original XR-Transformer work (Zhang et al., 2021a), the authors argued that one needs to perform clustering of the multi-label space in order to resolve scarce training instances in XMC. They also empirically showed that directly fine-tuning language models on extremely large output spaces is prohibitive. Furthermore, they empirically established that constructing clusters based on PIFA embedding with TFIDF features gives the best performance. However, no theoretical evidence was given in support of this approach to solving the XMC problem. We next leverage recent advances in graph learning to analytically characterize the benefits of using XR-Transformers.

*Description of the cSBM.* Using our Assumption 4.1, we analyze the case where the graph and node features are generated according to a cSBM (Deshpande et al., 2018) (see Figure 4). For simplicity, we use the most straightforward two-cluster cSBM. Let $\{y_i\}_{i=1}^n \in \{0,1\}$ be the labels of nodes in a graph. We denote the size of class $j \in \{0,1\}$ by $C_j = |\{i : y_i = j, \forall i \in [n]\}|$. We also assume that the classes are balanced, i.e., $C_0 = C_1 = \frac{n}{2}$. The node features $\{\mathbf{X}_i\}_{i=1}^n \in \mathbb{R}^d$ are independent $d$-dimensional Gaussian random vectors, such that $\mathbf{X}_i \sim N(\frac{r}{\sqrt{d}}\mathbf{1}, \frac{\sigma^2}{d}\mathbf{I})$ if $y_i = 0$ and $\mathbf{X}_i \sim N(-\frac{r}{\sqrt{d}}\mathbf{1}, \frac{\sigma^2}{d}\mathbf{I})$ if $y_i = 1$. The adjacency matrix of the cSBM is denoted by $\mathbf{A}$, and is clearly symmetric. All edges are drawn according to independent Bernoulli random variables, so that $\mathbf{A}_{ij} \sim Ber(p)$ if $y_i = y_j$ and $\mathbf{A}_{ij} \sim Ber(q)$ if $y_i \neq y_j$. Our analysis is inspired by Baranwal et al. (2021) and Li et al. (2019), albeit their definitions of graph convolutions and random walks differ from those in PIFA. For our subsequent analysis, we also make use of the following standard assumption and define the notion of *effect size*.

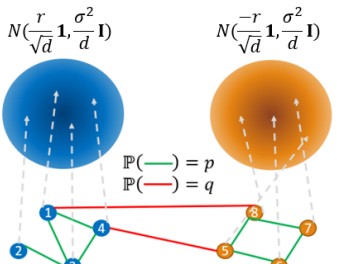

Figure 4: Illustration of a cSBM: Node features are independent Gaussian random vectors while edges are modeled as independent Bernoulli random variables.

**Assumption 4.2.** $p, q = \omega(\sqrt{\frac{\log n}{n}})$. $\frac{|p-q|}{p+q} = \Theta(1)$. $d = o(n)$. $0 < r, \sigma = \Theta(1)$.

Note that Baranwal et al. (2021) also poses constraints on $p, q, \frac{|p-q|}{p+q}$ and $d$. In contrast, we do not require $p - q > 0$ to hold (Baranwal et al., 2021; Li et al., 2019) so that we can address graph structures that are either homophilic or heterophilic. Due to the difference between PIFA and standard graph convolution, we require $p, q$ to be larger compared to the corresponding values used in Baranwal et al. (2021).

Table 1: Basic statistics of the OGB benchmark datasets (Hu et al., 2020a).

|  | #Nodes | #Edges | Avg. Node Degree | Split ratio (%) | Metric |
|---|---|---|---|---|---|
| ogbn-arxiv | 169,343 | 1,166,243 | 13.7 | 54/18/28 | Accuracy |
| ogbn-products | 2,449,029 | 61,859,140 | 50.5 | 8/2/90 | Accuracy |
| ogbn-papers100M | 111,059,956 | 1,615,685,872 | 29.1 | 78/8/14 | Accuracy |

**Definition 4.3.** *For cSBM, the effect size of the two centroids of the node features* $\mathbf{X}$ *of the two different classes is defined as*

$$\frac{\|\mathbb{E}\mathbf{X}_i - \mathbb{E}\mathbf{X}_j\|}{\sqrt{\mathbb{E}\|\mathbf{X}_i - \mathbb{E}\mathbf{X}_i\|^2} + \sqrt{\mathbb{E}\|\mathbf{X}_j - \mathbb{E}\mathbf{X}_j\|^2}}, \text{ where } y_i \neq y_j. \tag{3}$$

In the standard definition of effect size, the mean difference is divided by the standard deviation of a class, as the latter is assumed to be the same for both classes. We use the sum of both standard deviations to prevent any ambiguity in our definition. Note that for the case of isotropic Gaussian distributions, the larger the effect size the larger the separation of the two classes.

*Theoretical results.* We are now ready to state our main theoretical result, which asserts that the effect size of centroids for PIFA embeddings $\mathbf{Z}$ is asymptotically larger than that obtained from the original node features. Our Theorem 4.4 provides strong evidence that using PIFA in XR-Transformers offers improved clustering results and consequently, better feature quality.

**Theorem 4.4.** *For the cSBM and under Assumption 4.2, the effect size of the two centroids of the node features* $\mathbf{X}$ *of the two different classes is* $\frac{r}{\sigma} = \Theta(1)$. *Moreover, the effect size of the two centroids of the PIFA embedding* $\mathbf{Z}$ *of the two different classes, conditioned on an event of probability at least* $1 - O(\frac{1}{n^c})$ *for some constant* $c > 0$, *is* $\omega(1)$.

We see that although two nodes $i, j$ from the same class have the same neighborhood vectors in expectation, $\mathbb{E}\mathbf{A}_i = \mathbb{E}\mathbf{A}_j$, their Hamming distance can be large in practice. This finding is formally characterized in Proposition 4.5.

**Proposition 4.5.** *For the cSBM and under Assumption 4.2, the Hamming distance between* $\mathbf{A}_i$ *and* $\mathbf{A}_j$ *with* $y_i = y_j$ *is* $\omega(\sqrt{n \log n})$ *with probability at least* $1 - O(\frac{1}{n^c})$, *for some* $c > 0$.

Hence, directly using neighborhood vectors for self-supervision is not advisable. Our result also agrees with findings from the XMC literature (Chang et al., 2020b). It is also intuitively clear that averaging neighborhood vectors from the same class can reduce the variance, which is approximately performed by clustering based on node representations (in our case, via a PIFA embedding). This result establishes the importance of clustering within the XR-Transformer approach and for the SSL neighborhood prediction task.

## 5 EXPERIMENTS

**Evaluation Datasets.** We consider node classification as our downstream task and evaluate GIANT on three large-scale OGB datasets (Hu et al., 2020a) with available raw text: ogbn-arxiv, ogbn-products, and ogbn-papers100M. The parameters of these datasets are given in Table 1 and detailed descriptions are available in the Appendix E.1. Following the OGB benchmarking protocol, we report the average test accuracy and the corresponding standard deviation by repeating 3 runs of each downstream GNN model.

**Evaluation Protocol.** We refer to our actual implementation as GIANT-XRT since the multi-scale neighborhood prediction task in the proposed GIANT framework is solved by an XR-Transformer. In the pre-training stage, GIANT-XRT learns a raw text encoder by optimizing the self-supervised neighborhood prediction objective, and generates a fine-tuned node embedding for later stages. For the node classification downstream tasks, we input the node embeddings from GIANT-XRT into several different GNN models. One is the multi-layer perceptron (MLP), which does not use graph information. Two other methods are GraphSAGE (Hamilton et al., 2017), which we applied to ogbn-arxiv, and GraphSAINT (Zeng et al., 2020), which we used for ogbn-products as it allows for mini-batch training. Due to scalability issues, we used Simple Graph Convolution (SGC) (Wu et al., 2019) for ogbn-papers100M. We also tested the state-of-the-art GNN for each dataset.

Table 2: Results for the obgn-arxiv and ogbn-products datasets. Mean accuracy (%) $\pm$ one standard deviation. Boldfaced numbers indicate the best performances of downstream models, while underlined numbers indicate the best performance of models with a standard GNN pipeline for downstream models using $\mathbf{X}_{\text{plain}}$ and $\mathbf{X}_{\text{SSLGNN}}$. Methods under $\mathbf{X}_{\text{GIANT}}$ (GIANT framework) are part of the ablation study.

| Dataset | | ogbn-arxiv | | | | ogbn-products | | |
|---|---|---|---|---|---|---|---|---|
| | | MLP | GraphSAGE | RevGAT | RevGAT+SelfKD | MLP | GraphSAINT | SAGN+SLE |
| $\mathbf{X}_{\text{plain}}$ | OGB-feat† | 55.50 ± 0.23 | 71.49 ± 0.27 | 74.02 ± 0.18 | 74.26 ± 0.17 | 61.06 ± 0.08 | 79.08 ± 0.24 | 84.28 ± 0.14 |
| | BERT* | 62.91 ± 0.60 | 70.97 ± 0.33 | 73.59 ± 0.10 | 73.55 ± 0.41 | 60.90 ± 1.09 | 79.55 ± 0.85 | 83.11 ± 0.18 |
| $\mathbf{X}_{\text{SSLGNN}}$ | OGB-feat+GZ | 70.95 ± 0.38 | 71.41 ± 0.09 | 72.42 ± 0.16 | 72.50 ± 0.08 | 74.19 ± 0.55 | 78.38 ± 0.21 | 79.78 ± 0.11 |
| | BERT*+GZ | 70.46 ± 0.21 | 71.24 ± 0.19 | 72.33 ± 0.06 | 72.30 ± 0.20 | OOM | OOM | OOM |
| | OGB-feat+DGI | 56.02 ± 0.16 | 71.72 ± 0.26 | 73.48 ± 0.14 | 73.90 ± 0.26 | 70.54 ± 0.13 | 79.26 ± 0.16 | 81.59 ± 0.14 |
| | BERT*+DGI | 59.42 ± 0.38 | 72.15 ± 0.06 | 73.24 ± 0.25 | 73.60 ± 0.21 | 73.62 ± 0.23 | 81.29 ± 0.41 | 82.90 ± 0.21 |
| | OGB-feat+GAE | 56.47 ± 0.08 | 72.00 ± 0.27 | 73.70 ± 0.28 | 74.06 ± 0.10 | 74.81 ± 0.22 | 78.23 ± 0.10 | 82.85 ± 0.11 |
| | BERT*+GAE | 62.11 ± 0.32 | 72.72 ± 0.17 | 74.26 ± 0.20 | 74.48 ± 0.15 | 78.42 ± 0.14 | 82.74 ± 0.16 | 84.42 ± 0.04 |
| | OGB-feat+VGAE | 56.70 ± 0.20 | 72.04 ± 0.29 | 73.59 ± 0.17 | 73.95 ± 0.09 | 74.66 ± 0.10 | 78.65 ± 0.20 | 83.06 ± 0.06 |
| | BERT*+VGAE | 62.48 ± 0.14 | 72.92 ± 0.02 | 74.21 ± 0.01 | 74.44 ± 0.09 | 78.81 ± 0.25 | 82.80 ± 0.11 | 84.40 ± 0.09 |
| $\mathbf{X}_{\text{GIANT}}$ | BERT+LP | 67.33 ± 0.54 | 66.61 ± 2.86 | 75.50 ± 0.11 | 75.75 ± 0.04 | 73.83 ± 0.06 | 81.66 ± 0.08 | 82.33 ± 0.16 |
| | NO TFIDF+ NO PIFA | 69.33 ± 0.19 | 73.41 ± 0.34 | 74.95 ± 0.07 | 75.16 ± 0.06 | 74.16 ± 0.22 | 80.70 ± 0.51 | 81.63 ± 0.28 |
| | NO TFIDF+PIFA | 72.74 ± 0.17 | 74.43 ± 0.20 | 75.88 ± 0.05 | 76.06 ± 0.02 | 78.91 ± 0.28 | 81.54 ± 0.14 | 82.22 ± 0.15 |
| | TFIDF+NO PIFA | 71.74 ± 0.15 | 74.09 ± 0.33 | 75.56 ± 0.09 | 75.85 ± 0.05 | 79.37 ± 0.15 | 83.83 ± 0.14 | 85.01 ± 0.10 |
| | GIANT-XRT | **73.08 ± 0.06** | **74.59 ± 0.28** | **75.96 ± 0.09** | **76.12 ± 0.16** | **79.82 ± 0.07** | **84.40 ± 0.17** | **85.47 ± 0.29** |

At the time we conducted the main experiments (07/01/2021), the top-ranked model for ogbn-arxiv was RevGAT[2] (Li et al., 2021) and the top-ranked model for ogbn-products was SAGN[3] (Sun & Wu, 2021). When we conducted the experiment on ogbn-papers100M (09/10/2021), the top-ranked model for ogbn-papers100M was GAMLP[4] (Zhang et al., 2021b) (Since then, the highest reported accuracy was improved by 0.05% for ogbn-arxiv and 0.31% for ogbn-products; both of these improvements fall short compared to those offered by GIANT). For all evaluations, we use publicly available implementations of the GNNs. For RevGAT, we report the performance of the model with and without self knowledge distillation; the former setting is henceforth referred to as +SelfKD. For SAGN, we report results with the self-label-enhanced (SLE) feature, and denote them by SAGN+SLE. For GAMLP, we report results with and without Reliable Label Utilization (RLU); the former is denoted as GAMLP+RLU.

**SSL GNN Competing Methods.** We compare GIANT-XRT to methods that rely on graph-agnostic feature inputs and use node embeddings generated by various SSL GNNs modules. The graph-agnostic features are either default features available from the OGB datasets (denoted by OGB-feat) or obtained from plain BERT embeddings (without fine-tuning) generated from raw text (denoted by BERT*). For OGB-feat combined with downstream GNN methods, we report the results from the OGB leaderboard (and denote them by †). For the SSL GNNs modules, we test three frequently-used methods: (Variantional) Graph AutoEncoders (Kipf & Welling, 2016) (denoted by (V)GAE); Deep Graph Infomax (Velickovic et al., 2019) (denoted by DGI); and GraphZoom (Deng et al., 2020) (denoted by GZ). The hyper-parameters of SSL GNNs modules are given in the Appendix E.2. For all reported results, we use $\mathbf{X}_{\text{plain}}$, $\mathbf{X}_{\text{SSLGNN}}$ and $\mathbf{X}_{\text{GIANT}}$ (c.f. Figure 1) to denote which framework the method belongs to. Note that $\mathbf{X}_{\text{GIANT}}$ refers to our approach. The implementation details and hyper-parameters of GIANT-XRT can be founded in the Appendix E.3.

## 5.1 MAIN RESULTS

The results for the ogbn-arxiv and ogbn-products datasets are listed in Table 2. Our GIANT-XRT approach gives the best results for both datasets and all downstream models. It improves the accuracy of the top-ranked OGB leaderboard models by a large margin: 1.86% on ogbn-arxiv and 1.19% on ogbn-products. Using graph-agnostic BERT embeddings does not necessarily lead to good results (see the first two rows in Table 2). This shows that the improvement of our method is not merely

---

[2] https://github.com/lightaime/deep_gcns_torch/tree/master/examples/ogb_eff/ogbn_arxiv_dgl
[3] https://github.com/skepsun/SAGN_with_SLE
[4] https://github.com/PKU-DAIR/GAMLP

Table 3: Results for the obgn-papers100M dataset. Mean accuracy (%) $\pm$ one standard deviation. Boldfaced values indicate the best performance amongst the tested downstream models.

| ogbn-papers100M | | MLP | SGC | GAMLP | GAMLP+RLU |
|---|---|---|---|---|---|
| $\mathbf{X}_{\text{plain}}$ | OGB-feat[†] | $47.24 \pm 0.31$ | $63.29 \pm 0.19$ | $67.71 \pm 0.20$ | $68.25 \pm 0.19$ |
| | BERT[⋆] | $47.24 \pm 0.39$ | $61.69 \pm 0.29$ | $66.25 \pm 0.05$ | $67.15 \pm 0.07$ |
| $\mathbf{X}_{\text{GIANT}}$ | GIANT-XRT | $\mathbf{61.10 \pm 0.19}$ | $\mathbf{66.10 \pm 0.13}$ | $\mathbf{69.16 \pm 0.08}$ | $\mathbf{69.67 \pm 0.04}$ |

due to the use of a more powerful language model, and establishes the need for self-supervision governed by graph information. Another observation is that among possible combinations involving a standard GNN pipeline with a SSL module, BERT+(V)GAE offers the best performance. This can be attributed to exploiting the correlation between *numerical* node features and the graph structure, albeit in a two-stage approach within the standard GNN pipeline. The most important finding is that using node features generated by GIANT-XRT leads to consistent and significant improvements in the accuracy of all tested methods, when compared to the standard GNN pipeline: In particular, on ogbn-arxiv, the improvement equals $17.58\%$ for MLP and $3.1\%$ for GraphSAGE; on ogbn-products, the improvement equals $18.76\%$ for MLP and $5.32\%$ for GraphSAINT. Figure 5 in the Appendix E.6 further illustrate the gain obtained by our GIANT-XRT over SOTA methods on OGB leaderboard.

Another important observation is that GIANT-XRT is highly scalable, which can be clearly observed on the example of the ogbn-papers100M dataset, for which the results are shown in Table 3. In particular, GIANT-XRT improves the accuracy of the top-ranked model, GAMLP-RLU, by a margin of $1.42\%$. Furthermore, GIANT-XRT again consistently improves all tested downstream methods on the ogbn-papers100M dataset. As a final remark, we surprisingly find that combining MLP with GIANT-XRT greatly improves the performance of the former learner on all datasets. It becomes just slightly worse then GIANT-XRT+GNNs and can even outperform the GraphSAGE and GraphSAINT methods with default OGB features on ogbn-arxiv and ogbn-products datasets. This is yet another positive property of GIANT, since MLPs are low-complexity and more easily implementable than other GNNs.

## 5.2 ABLATION STUDY

We also conduct an ablation study of the GIANT framework to determine the relevance of each module involved. The first step is to consider alternatives to the proposed multi-scale neighborhood prediction task: In this case, we fine-tune BERT with a SSL link prediction approach, which we for simplicity refer to as BERT+LP. In addition, we examine how the PIFA embedding affects the performance of GIANT-XRT and how more informative node features (TFIDF) can improve the clustering steps. First, recall that in GIANT-XRT, we use TFIDF features from raw text to construct PIFA embeddings. We subsequently use the term "NO TFIDF" to indicate that we replaced the TFIDF feature matrix by an identity matrix, which contain no raw text information. The term "TFIDF+NO PIFA" is used to refer to the setting where only raw text information (node attributes) is used to perform hierarchical clustering. Similarly, "NO TFIDF+PIFA" indicates that we only use normalized neighborhood vectors (graph structure) to construct the hierarchical clustering. If both node attributes and graph structure are ignore, the result is a random clustering. Nevertheless, we keep the same sizes of clusters at each level in the hierarchical clustering.

The results of the ablation study are listed under rows indexed by $\mathbf{X}_{\text{GIANT}}$ in Table 2 for ogbn-arxiv and ogbn-products datasets. They once again confirm that GIANT-XRT consistently outperforms other tested methods. For BERT+LP, we find that it has better performance on ogbn-arixv compared to that of the standard GNN pipeline but a worse performance on ogbn-products. This shows that using link prediction to fine-tune BERT in a self-supervised manner is not robust in general, and further strengthens the case for using neighborhood instead of link prediction. With respect to the ablation study of GIANT-XRT, we see that NO TFIDF+NO PIFA indeed gives the worst results. Using node attributes (TFIDF features) or graph information (PIFA) to construct the hierarchical clustering in GIANT-XRT leads to performance improvements that can be seen from Table 2. Nevertheless, combining both as done in GIANT-XRT gives the best results. Moreover, one can observe that using PIFA always gives better results compared to the case when PIFA is not used. It aligns with our theoretical analysis, which shows that PIFA embeddings lead to better hierarchical clusterings.

ACKNOWLEDGEMENT

The authors thank the support from Amazon and the Amazon Post Internship Fellowship. Cho-Jui Hsieh is partially supported by NSF under IIS-2008173 and IIS-2048280. This work was funded in part by the NSF grant 1956384.

## 6 ETHICS STATEMENT

We are not aware of any potential ethical issues regarding our work.

## 7 REPRODUCIBILITY STATEMENT

We provide our code in the supplementary material along with an easy-to-follow description and package dependency for reproducibility. Our experimental setting is stated in Section 5 and details pertaining to hyperparameters and computational environment are described in the Appendix. All tested methods are integrated in our code: `https://github.com/amzn/pecos/tree/mainline/examples/giant-xrt`.

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

APPENDIX

## A    CONCLUSIONS

We introduced a novel self-supervised learning framework for graph-guided numerical node feature extraction from raw data, and evaluated it within multiple GNN pipelines. Our method, termed GIANT, for the first time successfully resolved the issue of graph-agnostic numerical feature extraction. We also described a new SSL task, neighborhood prediction, established a connection between the task and the XMC problem, and solved it using XR-Transformers. We also examined the theoretical properties of GIANT in order to evaluate the advantages of neighborhood prediction over standard link prediction, and to assess the benefits of using XR-Transformers. Our extensive numerical experiments, which showed that GIANT consistently improves state-of-the-art GNN models, were supplemented with an ablation study that aligns with our theoretical analysis.

## B    PROOF OF THEOREM 4.4

Throughout the proof, we use $\boldsymbol{\mu} = \frac{r}{d}\mathbf{1}$ to denote the mean of node feature from class $0$ and $\boldsymbol{\nu} = \frac{-r}{d}\mathbf{1}$ for class $1$. We choose to keep this notation to demonstrate that our setting on mean can be easily generalized. The choice of the high probability events will be clear in the proof.

The proof for the effect size of centroid for node feature $\mathbf{X}$ is quite straightforward from the Definition 4.3. By directly plugging in the mean and standard deviation, we have

$$\frac{2r}{2\sigma} = \Theta(1). \tag{4}$$

The last equality is due to our assumption that both $r, \sigma > 0$ are both constants.

To prove the effect size of centroid for PIFA embedding $\mathbf{Z}$, we need to first introduce some standard concentration results for sum of Bernoulli and Gaussian random variables.

**Lemma B.1** (Hoeffding's inequality (Hoeffding, 1994)). *Let $S_n = \sum_{i=1}^{n} X_n$, where $X_i$ are i.i.d. Bernoulli random variable with parameter $p$. Then for any $t > 0$, we have*

$$\mathbb{P}\left(|S_n - np| \geq t\right) \leq 2\exp(\frac{-2t^2}{n}). \tag{5}$$

**Lemma B.2** (Concentration of sum of Gaussian). *Let $S_n = \sum_{i=1}^{n} X_n$, where $X_i$ are i.i.d. Gaussian random variable with zero mean and standard deviation $\sigma$. Then for some constant $c > 0$, we have*

$$\mathbb{P}\left(|S_n| \geq c\sigma\sqrt{n\log n}\right) \leq 2\exp(-\frac{c^2}{2}\log n). \tag{6}$$

Now we are ready to prove our claim. Recall that the definition of PIFA embedding $\mathbf{Z}$ is as follows:

$$\mathbf{Z}_i = \frac{\mathbf{v}_i}{\|\mathbf{v}_i\|}, \text{ where } \mathbf{v}_i = \sum_{j:\mathbf{A}_{ij}=1} \mathbf{X}_j = [\mathbf{A}\mathbf{X}]_i. \tag{7}$$

We first focus on analyzing the vector $\mathbf{v}_i$. We denote $N_{iy} = |\{j : y_j = y \land \mathbf{A}_{ij} = 1, \ j \in [n]\}|$ and $N_i = N_{i0} + N_{i1}$. Without loss of generality, we assume $y_i = 0$. The conditional expectation of it is as follows:

$$\mathbb{E}\left[\mathbf{v}_i | \mathbf{A}\right] = N_{i0}\boldsymbol{\mu} + N_{i1}\boldsymbol{\nu}. \tag{8}$$

Next, by leveraging Lemma B.1, we have

$$\mathbb{P}(|N_{i1} - \frac{nq}{2}| \geq t) \leq 2\exp(\frac{-4t^2}{n}). \tag{9}$$

By choosing $t = c_1\sqrt{n\log n}$ for some constant $c_1 > 1$, we know that with probability at least $1 - O(\frac{1}{n^{c_1}})$, $N_{i1} \in [\frac{nq}{2} - c_1\sqrt{n\log n}, \frac{nq}{2} + c_1\sqrt{n\log n}]$. Finally, by our Assumption 4.2, we know that $nq = \omega(\sqrt{n\log n})$. Thus, we arrive to the conclusion that with probability at least $1 - O(\frac{1}{n^{c_1}})$, $N_{i1} = \frac{nq}{2} \pm o(1)$.

Following the same analysis, we can prove that with probability at least $1 - O(\frac{1}{n^{c_1}})$, $N_{i0} = \frac{np}{2} \pm o(1)$. The only difference is that we are dealing with $\frac{n}{2} - 1$ random variables in this case as there's no self-loop. Nevertheless, $(\frac{n}{2} - 1) = \frac{n}{2}(1 - o(1))$ so the result is the same. Note that we need to apply union bound over $2n$ error events ($\forall i \in [n]$ and 2 cases for $N_{i0}$ and $N_{i1}$ respectively). Together we know that the error probability is upper bounded by $O(\frac{1}{n^{c_2}})$ for some new constant $c_2 = c_1 - 1 > 0$. Hence, we characterize the mean of $\mathbf{v}_i$ on a high probability event $B_1$.

Next we need to analyze its norm. By direct analysis and condition on $\mathbf{A}$, we have

$$\|v_i\|^2 \stackrel{d}{=} \sum_{k=1}^{d}(\boldsymbol{\mu}_k N_{i0} + \boldsymbol{\nu}_k N_{i1} + \sum_{j=1}^{n} \mathbf{A}_{ij}\mathbf{G}_{jk})^2, \tag{10}$$

where $\mathbf{G}_{jk}$ are i.i.d. Gaussian random variables with zero mean, $\sigma$ standard deviation and $\stackrel{d}{=}$ stands for equal in distribution. Then by Lemma B.2 we know that with probability at least $1 - O(\frac{1}{N_i^{c_3}})$ for some constant $c_3 > 2 + c_2$

$$|\sum_{j=1}^{n} A_{ij}\mathbf{G}_{jk}| = O(\frac{\sigma}{\sqrt{d}}\sqrt{N_i \log N_i}). \tag{11}$$

This is because condition on $\mathbf{A}$, we are summing over $N_i$ Gaussian random variables. Recall that condition on our high probability event $B_1$, $N_i = \frac{n}{2}(p + q)(1 + o(1)) = \omega(\sqrt{n \log n}) \leq n$. Thus, we know that for some $c_2 > 0$, with probability at least $1 - O(\frac{1}{n^{c_2}} + \frac{1}{n^{c_3}}) = 1 - O(\frac{1}{n^{c_2}})$ we have $|\sum_{j=1}^{n} \mathbf{A}_{ij}\mathbf{G}_{jk}| = O(\frac{\sigma}{\sqrt{d}}\sqrt{n \log n})$. Again, recall that both $N_{i0}, N_{i1} = \omega(\sqrt{n \log n})$, thus together we have

$$\|v_{ik}\|^2 = \frac{n^2}{4}(\boldsymbol{\mu}_k p + \boldsymbol{\nu}_k q)^2(1 + o(1)). \tag{12}$$

$$\Rightarrow \|v_i\| = \frac{n}{2}\sqrt{\sum_{k=1}^{d}(\boldsymbol{\mu}_k p + \boldsymbol{\nu}_k q)^2(1 + o(1))} \tag{13}$$

$$= \frac{n}{2}|p - q|r(1 + o(1)). \tag{14}$$

Again, we need to apply union bound over $nd$ error events, which result in the error probability $O(\frac{1}{n^{c_2}})$ since $c_3 > 2 + c_2$ and $d = o(n)$ in our Assumption 4.2. We denote the corresponding high probability event to be $B_2$. Note that the same analysis can be applied to the case $y_i = 1$, where the result for the norm is the same and the result for $v_i$ would be just swapping $p$ and $q$. Combine all the current result, we know that with probability at least $1 - O(\frac{1}{n^{c_2}})$ for some $c_2 > 0$, the PIFA embedding $\mathbf{Z}_i$ equals to the following

$$\mathbf{Z}_i = \frac{\frac{n}{2}(\boldsymbol{\mu}p + \boldsymbol{\nu}q)(1 + o(1))}{\frac{n}{2}|p - q|r(1 + o(1))} = \frac{(\boldsymbol{\mu}p + \boldsymbol{\nu}q)(1 + o(1))}{|p - q|r} \text{ if } y_i = 0. \tag{15}$$

$$\mathbf{Z}_i = \frac{\frac{n}{2}(\boldsymbol{\mu}q + \boldsymbol{\nu}p)(1 + o(1))}{\frac{n}{2}|p - q|r(1 + o(1))} = \frac{(\boldsymbol{\mu}q + \boldsymbol{\nu}p)(1 + o(1))}{|p - q|r} \text{ if } y_i = 1. \tag{16}$$

Hence, the centroid distance would be

$$= \left\|\frac{(\boldsymbol{\mu}(p - q) + \boldsymbol{\nu}(q - p))(1 + o(1))}{r|p - q|(1 + o(1))}\right\| \tag{17}$$

$$= \left\|\frac{(\boldsymbol{\mu} - \boldsymbol{\nu})(p - q)(1 + o(1))}{r|p - q|(1 + o(1))}\right\| \tag{18}$$

$$= \frac{\|\boldsymbol{\mu} - \boldsymbol{\nu}\|}{r}(1 + o(1)) = 2(1 + o(1)). \tag{19}$$

Now we turn to the standard deviation part. Specifically, we will characterize the following quantity (again, recall that we assume $y_i = 0$ w.l.o.g.).

$$\left\|z_i - \frac{(\boldsymbol{\mu}p + \boldsymbol{\nu}q)}{r|p - q|}\right\| \tag{20}$$

Recall that the latter part is the centroid for nodes with label 0. Hence, by characterize this quantity we can understand the deviation of PIFA embedding around its centroid. From the analysis above, we know that given $\mathbf{A}$, we have

$$\left\| z_i - \frac{(\boldsymbol{\mu}p + \boldsymbol{\nu}q)}{r|p-q|} \right\| \leq \left\| \frac{N_{i0}\boldsymbol{\mu} + N_{i1}\boldsymbol{\nu}}{\|v_i\|} - \frac{(\boldsymbol{\mu}p + \boldsymbol{\nu}q)}{r|p-q|} \right\| + \left\| \frac{\sum_{j=1}^n \mathbf{A}_{ij}\mathbf{G}_j}{\|v_i\|} \right\|. \tag{21}$$

For the terms $\|v_i\|, N_{i0}, N_{i1}$ and $\|\sum_{j=1}^n \mathbf{A}_{ij}Z_j\|$, we already derive their concentration results above. Plug in those results, the first term becomes

$$\|\frac{N_{i0}\boldsymbol{\mu} + N_{i1}\boldsymbol{\nu}}{\|v_i\|} - \frac{(\boldsymbol{\mu}p + \boldsymbol{\nu}q)}{r|p-q|}\| \tag{22}$$

$$= \|\frac{\frac{n}{2}(\boldsymbol{\mu}p(1+o(1)) + \boldsymbol{\nu}q(1+o(1))}{\frac{n}{2}r|p-q|(1+o(1))} - \frac{(\boldsymbol{\mu}p + \boldsymbol{\nu}q)}{r|p-q|}\| \tag{23}$$

$$= \|\frac{(\boldsymbol{\mu}po(1) + \boldsymbol{\nu}qo(1))}{r|p-q|(1+o(1))}\| = \frac{r|p-q|o(1)}{r|p-q|(1+o(1))} = o(1). \tag{24}$$

The second term becomes

$$\|\frac{\sum_{j=1}^n \mathbf{A}_{ij}\mathbf{G}_j}{\|v_i\|}\| = \frac{O(\sigma\sqrt{n\log n})}{\frac{n}{2}r|p-q|(1+o(1))} \tag{25}$$

$$= \frac{O(\sigma\sqrt{n\log n})}{\omega(r\sqrt{n\log n})} = o(1), \tag{26}$$

where the last equality is from our Assumption 4.2 that $r, \sigma$ are constants. Together we show that the deviation of nodes from their centroid is of scale $o(1)$. The similar result holds for the case $y_i = 1$. Together we have shown that the standard deviation of $\mathbf{Z_i}$ is $o(1)$ on the high probability event $B_1 \cap B_2$. Hence, the effect size for the PIFA embedding is $\omega(1)$ with probability at least $1 - O(\frac{1}{n^{c_2}})$ for some constant $c_2 > 0$, which implies that PIFA embedding gives a better clustered node representation. Thus, it is preferable to use PIFA embedding and we complete the proof.

## C    PROOF OF PROPOSITION 4.5

Note that under the setting of cSBM and the Assumption 4.2, the Hamming distance of $\mathbf{A}_i, \mathbf{A}_j$ for $y_i = y_j$ is a Poisson-Binomial random variable. More precisely, note that

$$\forall k \in [n] \setminus \{i, j\} \text{ s.t.} y_k = y_i, \ |\mathbf{A}_{ik} - \mathbf{A}_{jk}| \sim Ber(2p(1-p)). \tag{27}$$

$$\forall k \in [n] \setminus \{i, j\} \text{ s.t.} y_k \neq y_i, \ |\mathbf{A}_{ik} - \mathbf{A}_{jk}| \sim Ber(2q(1-q)), \tag{28}$$

where they are all independent. Hence, we have

$$\text{Hamming}(\mathbf{A}_i, \mathbf{A}_j) \sim Bin(\frac{n}{2} - 2, 2p(1-p)) + Bin(\frac{n}{2}, 2q(1-q)), \tag{29}$$

where $\text{Hamming}(\mathbf{A}_i, \mathbf{A}_j)$ denotes the Hamming distance of $\mathbf{A}_i, \mathbf{A}_j$ and $Bin(a, b)$ stands for the Binomial random variable with $a$ trials and the success probability is $b$. By leveraging the Lemma B.1, we know that for a random variable $X \sim Bin(\frac{n}{2}, 2q(1-q))$, we have

$$\mathbb{P}\left(|X - nq(1-q)| \geq t\right) \leq 2\exp(\frac{-4t^2}{n}). \tag{30}$$

Note that the function $q(1-q)$ is monotonic increasing for $q \in [0, \frac{1}{2}]$ and has maximum at $q = \frac{1}{2}$. Combine with Assumption 4.2 we know that $nq(1-q) = \omega(\sqrt{n\log n})$. Hence, by choosing $t = \frac{\sqrt{cn\log n}}{2}$ for some constant $c > 0$, with probability at least $1 - O(\frac{1}{n^c})$ we have

$$X \geq nq(1-q) - \frac{\sqrt{cn\log n}}{2} = \omega(\sqrt{n\log n}). \tag{31}$$

Finally, by noting the fact that with probability 1 we have $Bin(\frac{n}{2} - 2, 2p(1-p)) \geq 0$. Hence, by showing $X \sim Bin(\frac{n}{2}, 2q(1-q))$ is of order $\omega(\sqrt{n\log n})$ with probability at least $1 - O(\frac{1}{n^c})$ implies that the Hamming distance of $\mathbf{A}_i, \mathbf{A}_j$ is of order $\omega(\sqrt{n\log n})$ with at least the same probability. Together we complete the proof.

# D PROOF OF LEMMA B.2

By Chernoff bound, we have

$$\mathbb{P}\left(S_n \geq a\right) \leq \min_{t>0} \exp(-ta)\mathbb{E}\exp(tS_n). \tag{32}$$

By the i.i.d assumption, we have

$$\min_{t>0} \exp(-ta)\mathbb{E}\exp(tS_n) = \min_{t>0} \exp(-ta)(\mathbb{E}\exp(tX_1))^n. \tag{33}$$

Note that the moment generating function (MGF) of a zero-mean, $\sigma$ standard deviation Gaussian random variable is $\exp(\frac{\sigma^2 t^2}{2})$. Hence we have

$$\min_{t>0} \exp(-ta)(\mathbb{E}\exp(tX_1))^n = \min_{t>0} \exp(\frac{1}{2}n\sigma^2 t^2 - ta). \tag{34}$$

By minimizing the upper bound with respect to $t$, we can choose $t = \frac{a}{n\sigma^2}$. Plug in this choice of $t$ we have

$$\mathbb{P}\left(S_n \geq a\right) \leq \exp(\frac{-a^2}{2n\sigma^2}). \tag{35}$$

Finally, by choosing $a = c\sigma\sqrt{n\log n}$ for some $c > 0$, applying the same bound for the other side and the union bound, we complete the proof.

# E EXPERIMENTAL DETAIL

## E.1 DATASETS

In this work, we choose node classification as our downstream task to focus. We conduct experiments on three large-scale datasets, ogbn-arxiv, ogbn-products and ogbn-papers100M as these are the only three datasets with raw text available in OGB. Ogbn-arxiv and ogbn-papers100M (Wang et al., 2020; Hu et al., 2020a) are citation networks where each node represents a paper. The corresponding input raw text consists of titles and abstracts and the node labels are the primary categories of the papers. Ogbn-products (Chiang et al., 2019; Hu et al., 2020a) is an Amazon co-purchase network where each node represents a product. The corresponding input raw text consists of titles and descriptions of products. The node labels are categories of products.

## E.2 HYPER-PARAMETERS OF SSL GNN MODULES

The implementation of (V)GAE and DGI are taken from Pytorch Geometric Library (Fey & Lenssen, 2019). Note that due to the GPU memory constraint, we adopt GraphSAINT (Zeng et al., 2020) to (V)GAE and DGI for ogbn-products. GraphZoom is directly taken from the official repository[5]. For all downstream GNNs in the experiment, we average the results over three independent runs. The only exception is OGB-feat + downstream GNNs, where we directly take the results from OGB leaderboard. Note that we also try to repeat the experiment of OGB-feat + downstream GNNs, where the accuracy is similar to the one reported on the leaderboard. To prevent confusion we decide to take the results from OGB leaderboard for comparison. For the BERT model used throughout the paper, we use "bert-base-uncased" downloaded from HuggingFace [6]. For the methods used in the SSL GNNs module, we try our best to follow the default setting. We slightly optimize some hyperparameters (such as learning rate, max epochs...etc) to ensure the training process converge. To ensure the fair comparison, we fix the output dimension for all SSL GNNs as 768 which is the same as bert-base-uncased and XR-Transformer.

## E.3 HYPER-PARAMETERS OF GIANT-XRT AND BERT+LP

**Pre-training of GIANT-XRT.** In Table 4, we outline the pre-training hyper-parameter of GIANT-XRT for all three OGB benchmark datasets. We mostly follow the convention of XR-Transformer (Zhang et al., 2021a) to set the hyper-parameters. For ogbn-arxiv and ogbn-products

---

[5]https://github.com/cornell-zhang/GraphZoom
[6]https://huggingface.co/bert-base-uncased

datasets, we use the full graph adjacency matrix as the XMC instance-to-label matrix $\mathbf{Y} \in \{0,1\}^{n \times n}$, where $n$ is number of nodes in the graph. For ogbn-papers100M, we sub-sample 50M (out of 111M) most important nodes based on page rank scores of the bipartite graph (He et al., 2016). The resulting XMC instance-to-label matrix $\mathbf{Y}$ has 50.0M of rows, 49.9M of columns, and 2.5B of edges. The PIFA node embedding for hierarchical clustering is constructed by aggregating its neighborhood nodes' TF-IDF features. Specifically, the PIFA node embedding is a 4.2M high-dimensional sparse vector, consisting of 1M of word unigram, 3M of word bigram, and 200K of character triagram. Finally, for ogbn-arxiv and ogbn-products, we use $TFN+MAN$ negative sampling to pre-train XR-Transformer where the model-aware negatives (MAN) are selected from top 20 model's predictions. Because of the extreme scale of ogbn-papers100M, we consider $TFN$ only to avoid excessive CPU memory consumption on the GPU machine.

Table 4: Hyper-parameters of GIANT-XRT. $HLT$ defines the structures of the hierarchical label trees. $lr_{max}$ is the maximum learning rate in pre-training. $n_{step}$ is the number of optimization steps for each layer of HLT, respectively. $B$ is total number of batch size when using 8 Nvidia V100 GPUs. $NS$ is the negative sampling strategy in XR-Transformer. $D$ is the $D$th layer of $HLT$ where we take the Transformer encoder to generate node embeddings as input for downstream GNN models.

| Dataset | $HLT$ | $lr_{max}$ | $n_{step}$ | $B$ | $NS$ | $D$ |
|---|---|---|---|---|---|---|
| ogbn-arxiv | 32-128-512-2048 | $6 \times 10^{-5}$ | 2,000 | 256 | $TFN+MAN$ | 4 |
| ogbn-products | 128-512-2048-8192-32768 | $6 \times 10^{-5}$ | 10,000 | 256 | $TFN+MAN$ | 2 |
| ogbn-papers100M | 128-2048-32768 | $6 \times 10^{-5}$ | 100,000 | 512 | $TFN$ | 3 |

**Pre-training on BERT+LP.** To verify the effectiveness of multi-scale neighborhood prediction loss, we consider learning a Siamese BERT encoder with the alternative Link Prediction loss for pre-training, hence the name BERT+LP. We implement BERT+LP baseline with the triplet loss (Balntas et al., 2016) as we empirically observed it has better performance than other loss functions for the link prediction task. We sample one positive pair and one negative pair for each node as training samples for each epoch, and train the model until the loss is converged.

### E.4 Hyper-parameters of downstream methods

For the downstream models, we optimize the learning rate within $\{0.01, 0.001\}$ for all models. For MLP, GraphSAGE and GraphSAINT, we optimize the number of layers within $\{1, 3\}$. For RevGAT, we keep the hyperparameter choice the same as default. For SAGN, we also optimize the number of layers within $\{1, 3\}$. For GAMLP, we directly adopt the setting from the official implementation. Note that all hyperparameter tuning applies for all pre-trained node features ($\mathbf{X}_{\text{plain}}$, $\mathbf{X}_{\text{SSLGNN}}$ and $\mathbf{X}_{\text{GIANT}}$).

### E.5 Computational Environment

All experiments are conducted on the AWS p3dn.24xlarge instance, consisting of 96 Intel Xeon CPUs with 768 GB of RAM and 8 Nvidia V100 GPUs with 32 GB of memory each.

### E.6 Illustration on the improvement of GIANT-XRT

See Figure 5.

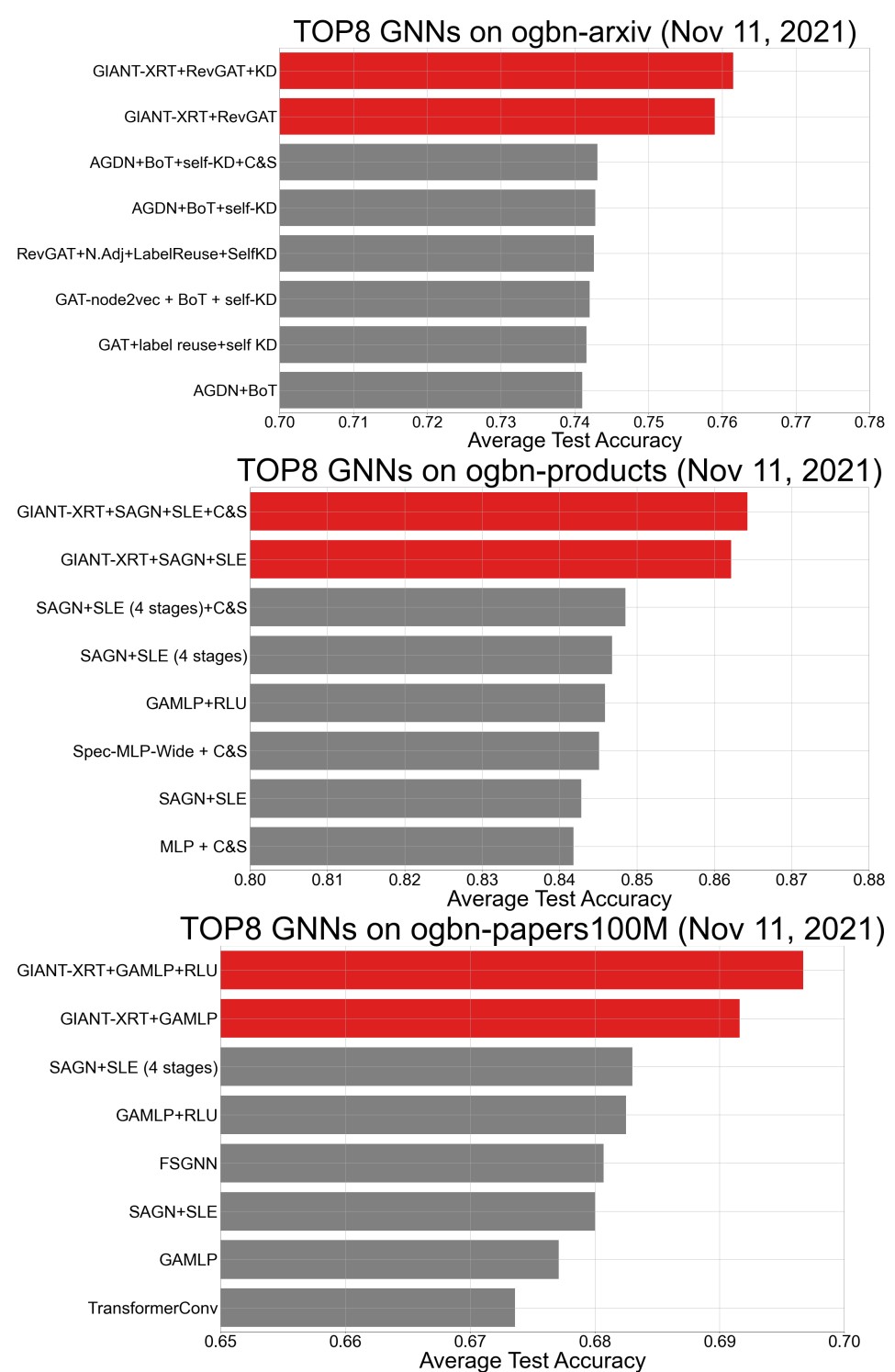

Figure 5: To further demonstrate that our GIANT-XRT indeed achieves a significant improvement over state-of-the-art methods, we plot the performance of top 8 models on OGB leaderboard (As of Nov. 11th, 2021). Note that our results on ogbn-products is better than those reported in Table 2, since we adopt the latest choice of hyperparameters provided in SAGN GitHub repository.

