# OpenReview forum: "Node Feature Extraction by Self-Supervised Multi-scale Neighborhood Prediction"
_ICLR.cc/2022/Conference — ICLR 2022 Poster_

### Official Review · Reviewer_3rZe · 2021-10-31

**Correctness:** 4
**Technical Novelty And Significance:** 3
**Empirical Novelty And Significance:** 3
**Recommendation:** 8
**Confidence:** 3

**Main Review:**

Strengths:
1. The problem is well motivated.
2. The proposed framework could be useful in general.
3. Both theoretical analysis and experiments are convincing.

Weaknesses:
1. the efficiency is not supported with experiments.
2. There are some typos.


**Summary Of The Paper:**

This paper develops a self-supervised learning framework to extract node features with the aid of graph. Connections between neighborhood prediction and the XMC problem are also established. Experiments on large-scale data show the superiority of the proposed method.

**Summary Of The Review:**

This paper develops a self-supervised learning framework to extract node features supervised graph. It is an interesting problem. Theoretical analysis is also provided. Extensive experiments on large-scale data validated the effectiveness of the proposed method.
There are some typos, e.g., "These have to be predicted using the textual information in order to best match the a priori given graph topology. This is achieved this by using the state-of-the-art XR-Transformer (Zhang et al., 2021a) method for
solving the XMC problem."

---

> ### Author Response · Authors · 2021-11-12
> **Response to Reviewer 3rZe**
>
> We thank reviewer 3rZe for their positive comments and valuable time. We address all concerns as follows.
>
> ### Q1: Efficiency of the experiments.
>
> Thanks for the great suggestion. We re-run the pre-training on ogbn-arxiv and ogbn-products to record the time complexity. However, since ogbn-papers100M is too large, it requires a special server to rerun the pre-training on it. This is unavailable at this moment. The running time of pre-training GIANT-XRT and downstream SOTA GNNs on all OGB datasets are as follows.
>
> **ogbn-arxiv**:
>
> - RevGAT+SelfKD : 1.5hr
> - GIANT-XRT: 1hr
>
> **ogbn-products**:
>
> - SAGN+SLE:3.5hr
> - GIANT-XRT:4hr
>
> We can see that the pre-training time of GIANT-XRT is roughly the same as downstream GNNs. It shows that our pre-training step is affordable in terms of time complexity. Note that we can pre-train GIANT-XRT just once and apply the resulting node features to many different downstream tasks (node labels), which is also the benefit of the pre-training procedure.
>
> ### Q2: Some Typos.
>
> Thanks for pointing out these typos. We will correct them in our revision.

---

> ### Author Response · Authors · 2021-11-25
> **Reply to Reviewer 3rZe**
>
> Dear Reviewer 3rZe,
>
> Thank you again for your valuable comments. We are wondering if your concerns have been addressed properly. Please let us know if you have any further questions after reviewing our answers.
>
> Best regards,
>
> The authors

---

> ### Comment · Reviewer_3rZe · 2021-11-25
> **I have read the rebuttal**
>
> The authors have addressed my concerns.

---

### Official Review · Reviewer_grtb · 2021-11-02

**Correctness:** 3
**Technical Novelty And Significance:** 4
**Empirical Novelty And Significance:** 3
**Recommendation:** 6
**Confidence:** 4

**Main Review:**


Strengths:
+ Introducing the idea of neighborhood prediction to guide self-supervised node feature learning is interesting and somewhat novel.
+ Connecting neighborhood prediction with the XMC problem is novel.
+ Extensive experiments are conducted on OGB and show new state-of-the-art results.


Weaknesses:
- The reviewer has some concerns on the provided theoretical analysis based on cSBM (Deshpande et al., 2018). It seems misleading and incomplete.

The theoretical analysis could be deduced from the analysis in Baranwal et al. (2021) with a few changes. In Baranwal et al. (2021), the cSBM is used to analysis the effect of graph convolution operation on the linear separability. The established theoretical results show that: if the means of the two mixture of Gaussians is not large than a threshold, the results after graph convolution are not guaranteed with high probability to improve the linear separability.

However, the statements in Theorem 4.4 is relatively vague. Note that PIFA is just one step of a graph convolution with the node features, plus a normalization step. What can we say about the performance of using the PIFA embedding?  Without the characteristic of the node features and the affinity of the graph convolution, it is hardly to have a convinsing conclusion.

Furthermore, the requirement on $p > q$, i.e., the probability $p$ of having a link between two nodes having the same label $y_i=y_j$ should be larger than the probability of having a wrong link between two nodes having different labels $y_i \neq y_j$. Is it necessary or not? Why?



**Summary Of The Paper:**

The paper proposed a self-supervised learning framework for learning node feature by exploring the correlation between the node feature and the graph structure, which leverages the graph information based on neighborhood prediction. To be specific, the proposed GIANT approach is combined with the pre-trained language model BERT, and incorporated the XMC formalism based on XR-Transformer. Partial theoretical analysis is also presented. Experiments conducted on three large benchmark datasets show promissing improvements.

**Summary Of The Review:**

The idea is clear and the empirical evaluation is strong. Since that the reviewer has some concerns on the provided theoretical analysis, it would be safe to decide after reading the feedback from the authors.

---

> ### Author Response · Authors · 2021-11-12
> **Response to Reviewer grtb**
>
> We thank reviewer grtb for their great comments and valuable time. We address all concerns as follows.
>
> ### Q1: Regarding the implication of Theorem 4.4.
> This is a very good question! We would like to emphasize that the goal of Baranwal et al. and ours are different, albeit they are somewhat related. As mentioned by Reviewer grtb, the main focus of Baranwal et al. is to show the linear separability threshold of the mixture of two Gaussians (and those after one-step graph convolution). Note that they focus on the semi-supervised setting of the problem, and it is more relevant to the downstream classification task. On the other hand, our goal is not to show linear separability. This is because we use PIFA to construct clusters of nodes and use the results as the self-supervised signal in XR-Transformer (See Figure 2). Hence, we aim to show that using PIFA gives us an embedding that has better clustering structures (based on cSBM and our Assumption 4.1) compare to not using PIFA. It is possible that even after using PIFA, the embedding is not linearly separable. However, since its effect size becomes smaller (our Theorem 4.4), it is still preferable to use PIFA embedding to construct clusters in XR-Transformer. Note that similar to the results in Baranwal et al. (2021), we also require both graph topology and node features to be informative enough. This is exactly stated in our Assumption 4.2, where $\frac{\|p-q\|}{p+q}=\Theta(1)$ corresponds to the part of graph information and $r,\sigma=\Theta(1)$ (and thus $\frac{r}{\sigma}=\Theta(1)$) corresponds to the part of node feature information.
>
> Note that it is hard to quantitatively characterize the exact relation between the final output quality of XR-Transformer and the clustering structure of PIFA embeddings. Nevertheless, we hope our analysis sheds light on the reason that PIFA embedding is beneficial in XR-Transformer, which is originally proposed in the XMC literature and empirically observed in Zhang et al. 2021a.
>
> ### Q2: Regarding the requirement on $p>q$.
>  Note that we do **not** require $p>q$. Only Li et al. 2019 and Baranwal et al. 2021 have this requirement. We have emphasized this point right below our Assumption 4.2. This is specifically important for our Assumption 4.1 since it implies our method does not bias toward homophily ($p>q$) or heterophily ($p<q$) cases. A similar discussion can also be found in Chien et al. 2021, where they suggested using cSBM to generate synthetic datasets for a different level of homophily/heterophily with $p>q$/$p<q$. It is unclear whether it is possible to extend the results of Li et al. 2019 and Baranwal et al. 2021 to the case $p<q$, which is not discussed by the original authors. Note that in the original cSBM paper
> (Deshpande et al. 2019), the information-theoretical recovery threshold does not depend on $p>q$ or $p<q$. This agrees with the classical SBM case [1].
>
> If one reads the arXiv preprint of Baranwal et al. (2021) carefully, they will find that only until Section 4.4, Lemma 3, Baranwal et al. emphasize the requirement of $p>q$. Section 4.4 is about the rate of decay of the classification loss, and we do not deal with linear separability. We conjecture that it is the linear separability proof that requires $p>q$. According to their paper and our analysis, the part convoluted result (their Section 4.3) does not need the assumption $p>q$. This is also the most relevant part to our case.
>
> **Remark:** To prevent potential misunderstanding, we decide to elaborate more on the requirement of $p>q$ in Baranwal et al. (2021). Since their Assumption 2 only requires $\frac{p-q}{p+q}=\Omega(1)$, one may think that $p<q$ is possible. However, they do state the requirement of $p>q$ in their proof. For example, the Lemma 3 in Section 4.4 of their arXiv preprint and the last paragraph in their Section 4.1 of their ICML version.
>
> ### Reference:
>
> [1] Community Detection and Stochastic Block Models, Abbe et al. JMLR 2017.

---

> ### Author Response · Authors · 2021-11-25
> **Reply to Reviewer grtb**
>
> Dear Reviewer grtb,
>
> Thank you again for your great comments. We are wondering if your concerns have been addressed properly. Please let us know if you have any further questions after reviewing our answers.
>
> Best regards,
>
> The authors

---

### Official Review · Reviewer_ZAh6 · 2021-11-08

**Correctness:** 1
**Technical Novelty And Significance:** 3
**Empirical Novelty And Significance:** 3
**Recommendation:** 6
**Confidence:** 4

**Main Review:**

Strengths: The proposed method is simple and reasonable. The experimental studies are extensive.

Weaknesses:

(1) The novelty is limited. In my view, masked language modeling aims to predict the masked tokens given the context, and in this paper, neighborhood prediction aims to predict the relation given the context. Relation-prediction-based objectives have been widely applied in knowledge/entity oriented pre-training, such as K-ADAPTER (Wang et al.),  ERICA (Qin et al.). In addition, the impact of the proposed method could be rather minor given their experimental results. For example, in Table 1, on the ogbn-arxiv dataset, based on GIANT-XRT, GraphSAGE, RevGAT, and RevGAT+SelfKD gain accuracies of 74.59%, 75.96%, and 76.12%, respectively, and based on TFIDF+NO PIFA, gain accuracies of 74.09%, 75.56%, and 75.85%. With such small gaps, it would be important to know whether the difference is actually statistically significant.

(2) Some details in Figure 1 are not clear, e.g., denotations of A and Y, full terms of XMC. To make Figure 1 self-contained, the caption of the figure should provide more necessary information.

(3) The idea are verified on three node classification datasets, while some details are missing. “Split ratio”in Table 1 is confusing, I cannot figure out what the ratios for train/test/development are. How many classes for each dataset? It would be better to show some real instances.

(4) What is the dataset used for pre-training GIANT?


**Summary Of The Paper:**

This paper presents a new self-supervised learning framework to enhance language model based on graph information.

**Summary Of The Review:**

The novelty is limited, and the impact of the proposed method could be rather minor. Some necessary details should be provided.

---

> ### Author Response · Authors · 2021-11-12
> **Response to Reviewer ZAh6 (1/2)**
>
> We thank reviewer ZAh6 for their comments and valuable time. We address all the concerns as follows.
>
> ### Q1: Novelty issue.
>
> We thank Reviewer ZAh6 brings up these two references on relation-prediction-based objectives. However, these two mentioned methods are actually more similar to **link prediction** in graph self-supervised learning literature, see our equation (2) for its mathematical formulation. For example, in ERICA the relation discrimination task (Section 3.4) uses cosine similarity as the similarity function of two representations (in their equation (3)). In K-ADAPTER, their Figure 4 in Appendix also shows that their pre-training tasks are more similar to link prediction as in ERICA. In contrast, our neighborhood prediction is different from link prediction, and we have a detailed discussion pertaining to their differences in Section 4. We show why such a similarity-based pre-training objective may not be suitable for graph data in some cases, where we give a simple counter-example in Figure 3. We have also conducted an ablation study in Section 5.2 and Table 2 (row of BERT+LP) to show that our neighborhood prediction task is indeed better than the link prediction task. Nevertheless, we will include these references in our revision.
>
> Here we want to emphasize the novelty of our work again. We are the first to propose extracting node features from raw data with the help of graph information to the best of our knowledge.  Furthermore, we are the first to bring the connection of the XMC problem and the newly recommended self-supervised learning objective (**multi-scale** neighborhood prediction) in the graph learning community. In contrast to the standard choice such as link prediction, we also provide an analysis of why our multiscale neighborhood prediction is better. This is also not previously discussed in the self-supervised learning GNN literature, see the nice literature review of Xie et al., 2021 cited in our paper. We hope Reviewer ZAh6 can correctly recognize the novelty of our work.
>
> ### Q2: Impact of GIANT.
> We respectfully disagree with the comment that the impact of our work is minor. In terms of the empirical results, note that we **consistently** improve the performance of all tested downstream methods on **all** OGB datasets. Especially, we achieve new state-of-the-art performance by a large margin on all three OGB datasets on the authentic OGB leaderboard. To make this point more clear, we add Figure 5 in Appendix D.6 to illustrate the gain obtained by our GIANT-XRT over state-of-the-art methods. We emphasize that our improvement is absolutely not minor.
>
> Note that TFIDF+NO PIFA is one of our ablation studies, and it belongs to our GIANT framework. We list this result to show that the components such as TFIDF and PIFA are both beneficial in XR-Transformer. This is clearly stated in Section 5.2. Our baseline methods should be those under the categories of $X_{plain}$ and $X_{SSLGNN}$. Indeed, our improvement over these methods is statistically significant (with at least 95% confidence based on the t-statistic).
>
> Besides the significant empirical improvement achieved by our GIANT-XRT on OGB datasets. We also want to highlight that our GIANT framework can be extended to other types of raw data formats, such as images and audio. This potential extension is also stated in the caption of Figure 1. Note that there is no prior work that investigates the problem of extracting node features from raw data with the help of graph information to the best of our knowledge. Yet, this is very important when dealing with real-world graph learning problems, especially when we demonstrate a huge improvement compared to baseline methods. We believe that our work can not only lead to a better solution in industrial applications but also initiate a new direction in both graph learning and XMC communities. Together, we hope Reviewer ZAh6 can reconsider the impact of our work.
>
> ### Q3: Regarding details of Figure 1.
> Thanks for the suggestion on improving the clarity of Figure 1. For the full terms of XMC, note that appears in the abstract (line 12-13). Hence, it should not be an issue. We agree that including a description of  $X$, $A$, and $Y$ is the caption can improve the clarity. Due to the space limit, we relegate these definitions to Section 3 in the current manuscript. We will try to include them in the caption of Figure 1 in our revision if there is enough space.

---

> > ### Author Response · Authors · 2021-11-12
> > **Response to Reviewer ZAh6 (2/2)**
> >
> > ### Q4: Regarding dataset statistics.
> >
> > Thanks for the suggestion. The split ratio refers to the train/validation/test split of node labels, where we follow the same terminology of the OGB paper [1]. There are 40/47/172 classes for ogbn-arxiv/ogbn-products/ogbn-papers100M respectively. More detailed statistics are provided in [OGB official site](https://ogb.stanford.edu/docs/nodeprop/) and also in their paper. We will include it in the appendix in our revision for clarity.
> >
> > ### Q5: ``What is the dataset used for pre-training GIANT?``.
> > We use the graph structure of OGB datasets as a self-supervised signal to pre-train our GIANT-XRT (illustrated in Figure 1 and thoroughly described by the whole Section 3). For example, for the experiment on ogbn-arxiv/ogbn-products/ogbn-papers100M, we use its graph structure and raw text attributes to pre-train GIANT-XRT. Then we use the generated node embedding, graph structures, and training labels of ogbn-arxiv to predict its test labels in downstream models. Regarding the BERT model used throughout the paper, we use “bert-base-uncased” downloaded from [HuggingFace](https://huggingface.co/bert-base-uncased).
> >
> > ### Reference
> > [1] Open Graph Benchmark: Datasets for Machine Learning on Graphs, Hu et al. Neural Information Processing Systems (NeurIPS), 2020.

---

> > > ### Comment · Reviewer_ZAh6 · 2021-11-12
> > > **Response to rebuttal**
> > >
> > > Thanks for the detailed interpretations. Most of my concerns have been solved, and I have improved my score. I hope you can provide the necessary details in your revised version.

---

### Decision · Program_Chairs · 2022-01-20

**Decision:**

Accept (Poster)

**Comment:**

In this submission, the authors presented a framework (GIANT) for self-supervised learning to improve LM by leveraging graph information. Reviewers agree that the method is somewhat novel, the (partial) theoretical analysis is interesting, and the evaluations are strong. We thank the authors for doing an excellent job in rebuttal which cleared essentially all the questions reviewers initially raised.